# Who belongs? Co-creating an assessment to measure belonging in a community space

**Monique Quinn**[1,2], **Sierra Roundtree**[3], **Lauren Porter**[4], **Kimberly Cutler**[4], **Kate Hanisian**[3], **Carley Riley**[4,5,6]*

**1** Division of Critical Care, Department of Pediatrics, Children's National Hospital, Washington, DC, **2** Department of Pediatrics, Division of Critical Care, Department of Pediatrics, George Washington School of Medicine and Health Sciences, Cincinnati Children's Hospital Medical Center, Cincinnati, Washington, DC, Ohio, **3** YMCA of Greater Cincinnati, Cincinnati, Ohio, **4** Michael Fisher Child Health Equity Center, Cincinnati Children's Hospital Medical Center, Cincinnati, Ohio, **5** Division of Critical Care, Department of Pediatrics, Cincinnati Children's Hospital and Medical Center, Cincinnati, Ohio, **6** Department of Pediatrics, University of Cincinnati College of Medicine, Cincinnati, Ohio

* Carley.Riley@cchmc.org

## Abstract

It is inherent to human nature to want to belong, and a strong sense of belonging has been closely correlated with multiple metrics of wellbeing on an individual and community level. Ensuring improvement in sense of belonging requires accurate measurement of belonging within the community of interest, an effort that has previously been constrained to primarily education and employment spaces and often lacking community voice. In this paper, we describe a novel approach to survey development, wherein we combine community voice with existent validated survey tools to develop a measure of a construct – sense of belonging in this instance – within a timeframe that better fits community desired pace for progress and change. In partnership with the YMCA of Greater Cincinnati, we worked within a Community Based Participatory Research (CBPR) approach to measure sense of belonging using a combination of literature review, extraction and qualitative research methods. External member checking and a Cronbach alpha analysis illustrate success in measuring the construct of interest: sense of belonging within the context of YMCA membership.

## Introduction

In 2023 the U.S. Surgeon General called attention to the negative health effects of loneliness and the power of social connection to mitigate these health effects [1]. Sense of belonging, a dynamic emotional attachment that relates individuals to others as well as the world they inhabit, has the potential to counteract loneliness and its negative impacts [34]. The link between sense of belonging and a myriad of positive impacts on health and wellbeing is well established. People with a high sense of belonging have better mental health compared to those with low belonging [2,3]. In fact, belonging better predicts physical health outcomes than many blood tests

**Data availability statement:** All relevant data are within the manuscript and its Supporting Information files.

**Funding:** The project team received a Partnership Development Grant for this work, through the NIH. The primary author on the grant was MQ, supported by all three other authors. The sponsor of the award played no role in the project beyond funding. The project described is supported by the National Center for Advancing Translational Sciences of the National Institutes of Health, under Award Number UL1TR001425. The content is solely the responsibility of the authors and does not necessarily represent the official views of the NIH.

**Competing interests:** The authors have declared that no competing interests exist.

[4–6]. High sense of belonging is associated with greater resilience, better memory, decreased effects of stress and trauma, stronger relationships, lower exposure to child abuse and neglect, lower risk of mortality, and greater social capital [7–11]. People with greater sense of belonging are more likely to ask for help when in need, advocate for themselves and their families, and provide aid to others within their communities [10].

With the large body of evidence supporting the important role that sense of belonging plays in wellbeing, it is unsurprising that many organizations and communities seeking to improve population wellbeing are attempting to do so through interventions focused on improving sense of belonging. Accurately measuring sense of belonging in each context is therefore important in determining success of those efforts over time. A 2022 narrative review on belonging conceptualization found that there are two main forms of belonging: "trait belongingness" (defined as a core psychological need that is stable over time) and "state belongingness" (which is situation specific belonging mediated by daily events, group dynamics, and stressors) [12]. State belongingness has become the primary focus for organizations interested in belonging within their communities. To that end, a plethora of survey tools primarily focused on state belongingness have already been created and validated within the contexts of education and employment spaces. It is unclear, however, if these existing survey tools are transferable to other contexts including community organizations. First, the processes to create these survey tools may not have centered community voice, which is important in ensuring that the researcher is measuring a construct as known and understood by the population. Second, we believe that each context and community is unique, so even those survey tools that have been developed with community input are not automatically relevant to new or other community contexts.

Community based participatory research (CBPR) has become a standard methodological approach used by researchers who recognize the importance of community engagement and voice in their work [13]. Positioning communities and their lived experience as equal experts in the research paradigm, CBPR is Intended to be a guide for shared power and decision making between communities and researchers, enriching both the process and outcomes of research. CBPR is lauded as an improvement from traditional research through both an equity and an ethics lens, and research developed in partnership with community has demonstrated manifold successes in sustained health improvements on an individual and community level. Despite its' many benefits, one of the drawbacks to utilizing any research approach for community change – including CBPR – is that the timeframe of research and the timeframe desired by community for action and results are often mismatched [14–16]. With this consideration in mind, efforts that engage community partners in research must continuously attend to the timeliness of their efforts, agree upon timelines that fit both invested parties, and seek methods that yield valid and replicable results in expedient ways.

In this study, we illustrate a replicable process whereby community voice can inform measurement instrument development without sacrificing forward momentum or time. Additional novelty of our work includes focusing on sense of belonging within

a community outside of the previously studied education and employment spaces. Our community engaged research was conducted within a community-academic partnership between the YMCA of Greater Cincinnati (YMCA) and the Mayerson Child Wellbeing Initiative (MCWI) at Cincinnati Children's Hospital Medical Center (CCHMC). The two groups share a vision of inclusive wellbeing in their community, with the YMCA identifying sense of belonging as an organizational priority for growth. Through the sharing of efforts and expertise, we combined qualitative research analysis with existent validated survey items on sense of belonging to yield a usable survey to track sense of belonging in the organization over time.

## Materials & methods

### Overview & partnership development

In 2023, the YMCA of Greater Cincinnati's strategic plan included a new focus on sense of belonging, with the recognition that sense of belonging is a crucial pillar of social connectedness and community formation. They sought partnership with MCWI due to shared goals of improving inclusive wellbeing and thriving. In alignment with CBPR strategies, this community-academic partnership co-created a project charter that included the joint vision, goals of the research, roles and responsibilities of the partners, and anticipated timeline for the project. The YMCA team contributed knowledge in strategic planning, business operations, collaborative leadership, and community engagement, while the MCWI team leant expertise in research methodology, rigor and output as well as in systems thinking. Both partners provided input and discourse on project steps and results throughout the project, with the YMCA leading pace and direction. CCHMC served as the IRB of record, with this study being approved under expedited review with the determination of no greater than minimal risk to participants.

### Existing literature

We conducted an extensive literature search to understand sense of belonging as a construct as well as how to measure it. We included articles that were available in English and discussed the development or evaluation of a measurement tool of belonging, regardless of when the article was published. We additionally reviewed bibliographies of articles that met our inclusion criteria, to identify other sources and surveys. In total, we found twenty validated assessment tools that measured at least one aspect of sense of belonging (state versus trait belongingness).

### Focus groups

After understanding the landscape of literature, we needed to know how Cincinnati YMCA members thought of belonging before we could accurately assess it. Specifically, we wanted to know about how members internalize belonging as a concept, including their perceptions on facilitators and barriers to belonging. We conversed with YMCA members through focus groups rather than individual interviews to hear from more people and to allow for richer and nuanced discussion. We utilized purposive sampling to recruit focus group members 12 years and older at a single YMCA branch selected for readiness for engagement as well as diversity of membership across age, race, and income level. The project team discussed the recruitment strategy which was then carried out by YMCA team members. We recruited through onsite signage, emails to the branch listserv, as well as direct outreach by staff at the front desk. We used purposive sampling to ensure focus groups closely matched the demographics of the branch. We targeted focus group size of 6–10 people. Recruitment and focus group participation occurred from 9/18/23–11/28/23.

We hypothesized that participation and discussion within each focus group would be richer when members were among peers with shared lived experiences (particularly as it relates to life stages commonly determined by age). We therefore sought to recruit focus groups by age through hosting focus groups at varying times of day where certain age groups were more likely to be present. No person was excluded if they attended a focus group outside of the target age range. Focus groups were held onsite at the local YMCA, for ease of access and familiarity. The branch offered childcare.

The research team also provided refreshments, and each participant received a $25 gift card to a local grocery store at the end of the focus group.

We conducted focus groups with the assistance of a semi-structured interview guide (S1 Fig). The MCWI research team created the interview guide, focusing on definitions and understanding of belonging on the individual and community level, and the YMCA project team provided feedback and adjustments prior to finalization of the guide. The same researcher from the MCWI team led all focus groups, reducing potential for variation. Participants received a written research information sheet that was also read out loud by the focus group leader at the beginning of the session. We obtained IRB approval for waiver of written consent due to the low-risk nature of this study. For focus group members who were under the age of 18, we requested that their parents be present for the introduction of the focus group. We additionally obtained IRB approval for waiver of written parental permission for child participants. Ongoing participation in the focus group served as a passive indicator of consent and assent for all participants, and we informed focus group members that they were free to leave at any time. Before beginning the session, we asked all participants to complete an anonymous demographic questionnaire. The focus group, and recording, started after all participants had turned in their questionnaire.

## Qualitative analysis

We approached this project through a constructivist paradigm, believing that every person has unique lived experiences that influence their conceptualization of sense of belonging. Focus groups continued serially until we reached thematic saturation. We followed Braun and Clarke's six phase thematic analysis framework to inductively identify patterns in experiences of belonging within the Cincinnati YMCA community [17]. Step 1 (familiarization of the data): We transcribed voice recording verbatim to written text using Microsoft Word free transcription process, and one researcher checked for accuracy and redacted all identifiers prior to further analysis. Two research team members (one from MCWI, one from YMCA) read through all transcripts before coding began until they felt comfortable with the data. They then proceeded with Step 2 (generating initial codes): separately using an open coding process to manually code each transcript. As part of Step 3 (searching for themes), the two researchers met after each focus group was coded individually to compare codes and group them into themes that resulted in the initial code book. Intentional time was taken at these meetings to examine personal and interpersonal reflexivity in the analytic process. Once they completed all analyses and had compiled a final code book, the coding pair then met with the rest of the project team to discuss and interpret findings as part of Step 4 (reviewing themes). This yielded further questions, clarifications, return to the qualitative data, and editing of the codebook. The two researchers involved in the data analysis had final say in editing the codebook and finalizing themes. Naming the themes (Step 5) was the last step in this process before comparing our codebook to existent surveys of belonging.

## Researcher reflexivity

The primary author of this paper led all focus groups and was one of two people who coded the data. She identifies as cisgender female, White, and is a physician with research interests in population wellbeing and thriving. She has had mentored and experiential learning in qualitative research work, including leading focus groups and qualitative data analysis. At the time of the study, she was actively in an educational certificate program in Community Engaged Research for Health that provided additional training in qualitative work, understanding and mitigating power dynamics, and community based participatory research. When leading focus groups, she did not disclose her occupation in an effort to reduce impressions of power from participants, and instead solely named herself as an employee at Cincinnati Children's Hospital. She was not a member of the YMCA at the time of this study. The second researcher who participated in data analysis was also the primary project manager as an employee at the YMCA. She identifies as a cisgender African American woman with a passion for community engagement that emphasizes co-creation. She led recruitment efforts for the project and had no prior relationships or connection with the individuals in the focus groups. She did not participate in the focus groups beyond recruitment in order to allow YMCA members to speak truthfully without fear of retribution and to reduce

 

concerns related to power imbalances. To further this goal, she did not review any focus group transcripts until after anonymization. In her job role at the YMCA as Innovation Project Manager she has extensive training in project leadership, inclusive group work, and community outreach. She received training and mentorship in qualitative data analysis by the primary author before undertaking coding in this project. Both researchers together reflected on their own perceptions of belonging during the data analysis process, including discussions of how that might impact their interpretation of the participants dialogue. The researcher employed at the YMCA also reflected on her sense of belonging at the YMCA and the influence that might have on her data interpretation.

### Survey creation & validation

Once qualitative analysis of focus group interviews was complete, validated survey questions were reviewed and compared to the identified themes. When multiple questions were identified as fitting a particular theme, the leadership team met and came to consensus about which wording had highest concordance with the meaning of the theme. After the initial novel survey tool was created, we conducted external member checking by presenting the survey to YMCA members at the same branch but who were not involved with the focus groups to assess construct validity. A research team member was present at the branch at varying time points of the day to capture different demographics of membership and approached individuals after check-in. Finally, the survey was distributed to primary account holders across the entire membership of the YMCA of Greater Cincinnati. Cronbach α statistical analysis was performed to determine internal consistency and reliability.

## Results

We held five focus groups, with a total of 30 participants. Focus group demographics closely matched that of the YMCA branch membership in terms of age and gender. When evaluating representation by race, the focus groups had an over-representation of participants who self-identified as Black and Asian compared to branch makeup (S1 Table). In regard to the literature review, we found twenty individual survey instruments that sought to measure one or more components of sense of belonging in a variety of populations. Ten of the validated instruments had at least one question that assessed belonging as understood by YMCA members and were referenced in our survey creation (see the Table embedded within each theme, or S2 Table for full map).

Within the focus groups, we observed that belonging spaces of participants differed based on age: youth focused on friend groups with shared interests, working adults focused on career and family spaces, and retired adults focused on recreational activities. That said, there were many shared reflections about belonging across the age spectrum. Thematic analysis revealed six major themes about belonging (see Supplemental Materials for full quote tables). Three of these themes – belonging requires reciprocity, the drive to belong is innate and immutable, and belonging is relational – matched validated belonging survey items in the literature and were used to develop a belonging assessment tool specific to this community. Three remaining themes (belonging is fostered through commonalities, the benefits of belonging, and belonging as synonymous with membership) were excluded from survey development for several reasons. The themes identified by this community but not represented well in existent academic literature provide useful insights into construct conceptualization at a membership-based organization.

### Theme 1: Belonging requires reciprocity

Focus group participants discussed how reciprocity is needed to facilitate belonging. They rejected any notion that belonging is transactional while asserting that both parties (the person seeking belonging and the space they were joining) needed to participate in order for the individual to experience a high sense of belonging. This reciprocity, or bidirectional engagement, was discussed in terms of effort from the individual: *"I think also [belonging means] to be actively involved. You know, you can belong to, be a part of something but not really belong. If you just are not active in […] what you're supposedly a part of."* (FG1, Speaker 8). Effort was also needed from the group a person was trying to join: *"When*

*someone at your safe space or your belonging space can genuinely reach out to you, and it just feels good. And they're, you know, just reaching out to you. Like 'oh, I have a good relationship. I belong with that person'. It's reciprocated." (FG3, Speaker 4).* In this way, focus group participants described the need for reciprocal effort to create a sense of belonging.

Reciprocity was also touched on as participants considered the importance of value to feel belonging. On the one hand, this included contributing value to a group: *"Because you want to bring value to sort of any situation that you're in. […] you know, you want to bring something to the table, even if it's just something little." (FG5, Speaker 5).* On the other hand, it was also important to feel valued by the group to which there is belonging: *"Consistently being overlooked and forgotten really makes you feel like a background character. Like you're simply there, but no one interacts with you. At that point the sense of belonging in community really starts to dissolve." (FG3, Speaker 2).* Value and effort being reciprocally felt within a belonging space was therefore a core tenet of sense of belonging in this group.

While reciprocity is not directly assessed in existent belonging measurement tools, the idea of feeling valued and actively included appears in several published survey items (Table 1).

Table 1. **Validated assessment of value as a component of belonging.** This table describes how existent survey instruments assess "Value", as a known component of belonging. Within the table, titles of the validated scales can be found in the left column, while the exact phrasing used by each scale can be found in the column to the right.

## Theme 2: The drive to belong is an innate part of being human

Every participant in the focus groups was familiar with the term 'sense of belonging' and the construct was described as an unanimously positive thing, *"Feeling happy, feeling part of a group. It's a positive word, not just. […] not just being an outcast in a group. Belong, you belong." (FG4, Speaker 2).* Focus group members felt that all human beings innately want to belong, and the drive to seek belonging spaces is part of what makes us human, *"People want to connect. I feel like it's kind of human nature, at the end of the day." (FG2, Speaker 6).* Although all individuals had their own nuanced interpretation of belonging, all agreed that it was part of the human experience and well known to them. Validated surveys in the literature drew on this construct familiarity by asking about belonging directly, as represented in Table 2.

Table 1. Validated assessment of value as a component of belonging.

| Existent Scale Name | Existent Survey Item Phrasing |
| --- | --- |
| Simple School Belonging Scale [18] | I feel like I matter to people at [school name]. |
| Belonging Barometer [19] | People in [name of respondent's local community] value me and my contributions.<br>People in [name of respondent's local community] welcome and include me in activities. |
| University Belonging Questionnaire [20] | I feel that a faculty member has valued my contributions in class. |
| SOBI-P* (reverse phrasing) [21] | I am not valued by or important to my friends. |

Table 2. Validated assessments asking about belonging directly due to its familiarity.

| Existent Scale Name | Existent Survey Item Phrasing |
| --- | --- |
| National Youth Outcomes Initiative [22] | I feel like I belong here. |
| Perceived Cohesion Scale [23] | I feel a sense of belonging to [blank]. |
| General Belonging Scale [24] | I have a sense of belonging. |
| Malleable Social-Psychological Academic Attitudes Survey [25] | I feel like I belong in my school. |
| National Study of Living-Learning Programs survey instrument [26] | I feel a sense of belonging. |
| New Brunswick School Climate Survey [27] | I feel like I belong at this school. |
| Belonging Barometer [19] | When interacting with people in [name of respondent's local community], I feel like I truly belong. |

Table 2. **Validated assessments asking about belonging directly due to its familiarity.** This table describes how existent survey instruments assess belonging directly. Within the table, titles of validated scales can be found in the left column, while the exact phrasing used by each scale can be found in the column to the right.

### Theme 3: Belonging is relational and combats loneliness

Although published literature on sense of belonging differentiates between connection, community, and belonging, focus group participants spoke about them interchangeably. For example, when asked what comes to mind when they hear the word 'belonging', participants had this to share: *"Companionship […] We're supposed to share and learn from each other. Companionship is very, as you get older, you know companionship is very important." (FG1, Speaker 3)* and *"I think [community and belonging] are pretty close. They are, I mean, they might not be the exact same, but that's pretty close to me." (FG5, Speaker 5)*. This relational tenet of belonging became important when participants compared spaces where they felt low and high belonging.

When considering spaces of high belonging, participants uniformly discussed spaces that were rich in authenticity and trust, such that they felt safe to be themselves, *"I feel like I belong to something when I have a sense of trust within, I feel like there's a non-judgmental place. I can be truthful and not feel judged." (FG3, Speaker 4)*. The existence of safe spaces is fostered through a deeper sense of knowing a person, *"To really like have a community, like a connection with someone, you gotta understand the good and bad about someone." (FG2, Speaker 6)*. This presence or absence of safety and trust became particularly important as participants considered how low belonging spaces are different: *"It's easy [to connect on social media], but it's not real." (FG2, unknown speaker)*; *"Or you try to hide your true colors to make the feeling of belonging, but it becomes a false sense […]" (FG3, Speaker 3)*. Thus it was evident that building relationships rich in belonging requires psychological safety and authenticity.

The relational component of sense of belonging was also evident in how participants spoke of belonging as an antidote to loneliness. In fact, multiple participants named the desire to avoid loneliness as a motivator for them to seek a space of belonging.

*"We find belonging, and you will even go to the point of being fake to find belonging, to combat loneliness I feel like at its core. Cuz no one wants to be alone. And if you even find that, and just like with a mask on, that little bit of like feeling of belonging can fight that feeling of loneliness." (FG3, Speaker 3).*

So while authenticity within relationships was crucial to high belonging, multiple participants named that inauthentic relationships were preferable to a lack of relationships altogether.

The relational aspects of belonging, including its' roots in high-trust spaces and its' ability to combat loneliness, is represented in existing surveys using the phrasing as seen in Table 3.

Table 3. **Validated assessments asking about a relational component of belonging.**

| Existent Scale Name | Survey Item Phrasing |
| --- | --- |
| University Belonging Questionnaire [20] | I feel connected to a faculty/staff member at my university. |
| General Belongingness Scale [24] | I feel accepted by others.<br>I feel connected with others. |
| Belonging Barometer [19]<br>*Reverse phrasing | I feel emotionally connected to [name of respondent's local community].<br>I feel unable to be my whole and authentic self with people in [name of respondent's local community].* |
| SOBI-P [21]<br>*Reverse phrasing | I often wonder if there is any place on earth where I really fit in* |

Table 3. **Validated assessments asking about a relational component of belonging.** This table describes how existent survey instruments assess relationality. Within the table, titles of validated scales can be found in the left column, while the exact phrasing used by each scale can be found in the column to the right.

## Theme 4: Sense of belonging is initiated and fostered through commonalities

We asked participants about what made them feel like they belonged more or less within a particular space. Repeatedly people discussed that having something in common heightened their sense of belonging going so far as to name that it was the foundation of belonging in many cases. This attention to common ground (or the lack thereof) was true for recreational activities: *"I joined a book club and all the women in my book club are under the age of 60 and I'm 72. […] But it's like, you know, we all have the same kind of feelings about life and things like that,"* (FG1, Speaker 9). The expression of sharing something with others is similarly seen in the job space: *"Everyone felt like, 'Oh, OK, we have purpose. We have goals.' We all kind of felt, I would say, a sense of belonging together."* (FG3, Speaker 2).

Not only were shared goals or interests helpful in fostering belonging when those experiences were positive, but participants also named that sharing an experience of adversity created deep bonds: *"I was on the Colerain basketball team last year and my best friends now are all the people who were on the team and some other people. But like, it's usually the people who I like shared blood sweat and tears with."* (FG2, Speaker 7). And, *"I belong to a grief group, online on Facebook that I joined after my son passed […] and we've all suffered the loss of our sons. And it's like a group that we have that common thread, and we can feel each other even though they are in different parts of the world, the country,"* (FG1, Speaker 7). Similarities or commonalities played a clear role in sense of belonging among this population, however we did not find representation of this concept in our literature review of surveys on sense of belonging and thus it was not included in our survey.

## Theme 5: The benefits of belonging are many and multifaceted

The focus groups were asked to consider the impact of belonging on both an individual and community level. Understanding impact was necessary for the research team to understand whether belonging is considered important to the community (and not just the research team). The question, and the manifold benefits described by community members in response to it, served to provide evidence for or against prioritizing belonging on an organizational level. As such, it was not intended to inform the measurement of sense of belonging. Indeed, the responses that the impact questions elicited described multilevel benefits (from the individual to the community) but thematic analysis did not reveal codes or themes that allowed for a deeper assessment of an individuals sense of belonging.

In terms of individual benefit, one of the most commonly discussed aspects was higher self-confidence: *"So the groups that I'm in, I always feel like it's something that I'm doing to better myself, to enhance my myself and my feeling my self-worth you know."* (FG1, Speaker 7). Participants also frequently discussed belonging contributing to self-growth as it relates to being part of a diverse group of people: *"Different ideals I mean are being plugged into the community so everybody can learn. […] That's what a community is, it's a group of people connecting, right […] I'm just learning from other people."* (FG2, Speaker 6). No participant commented on a negative impact of belonging at the individual level.

When considering the impact of belonging at the community level, the most frequently discussed topics included how every person contributes a unique perspective that enriches the whole community, that communities with high belonging are more likely to have members that give back to their community, and that high-belonging communities have increased recruitment and retention. In terms of unique perspectives enriching the community, one participant said, *"We all have different talents and gifts, and I think that's what would make, you know, a really strong solid group."* (FG1, Speaker 4).

Regarding giving back to one's community, participants named internal motivation beget from the positive feelings that belonging created, *"And I think when you feel like you belong it empowers you to want to do better or to help others."* *(FG3, Speaker 2)*; *"It's easy to go above and beyond and do more than what is asked when you have that kind of feeling."* *(FG5, Speaker 6).* Participants followed similar lines of thinking when discussing how the positive feelings associated with belonging make it more likely for a person to stay in a group and recruit others to join: *"If they see I join a new community that makes me feel good… if they notice that, they're going to too." (FG2, Speaker 2).* While not directly contributing to survey construction, we include this theme within our analysis as it illustrates that sense of belonging is an important construct to community and academic alike, and is considered beneficial by community members – warranting continued focus.

### Theme 6: Belonging is akin to membership

The last theme that arose from the data was the frequency with which focus group participants likened belonging to membership with a group. Insinuating synonymity between belonging and membership was seen when participants were asked to define belonging and used words like *"fitting in" (FG2, Speaker 2)* and *"to be a part of" (FG1, Speaker 3)* without further descriptors regarding the depth of connection that may or may not exist. It was also heard by participants when they considered what factors might foster belonging. On this topic, the majority of conversation across all focus groups centered on nonverbal cues *"You come in here and you're greeted with a smile. […] what's not to like?" (FG1, Speaker 5)* and creating a welcoming environment through greetings and acknowledgements, *"It could be simply not forgetting to say 'good morning' or 'hi' to you every single day that you see them. Not just walking by, but being recognized in that space can mean a lot to a lot of people." (FG3, Speaker 3).* The fact that belonging was felt by focus group participants through greetings and acknowledgements juxtaposes the prior theme of belonging being rooted in trust and authentic knowing of another. This conceptualization of belonging was most often seen when participants discussed belonging within the YMCA, particularly describing times where they felt an affinity or sense of membership to the organization.

Although this theme was identified with regularity across the qualitative data, the decision was made to not consider it for survey development for several reasons. The first is its conflict with Themes 1 & 3, where bidirectional engagement is required for rich belonging, and belonging spaces require trust and authenticity. Commentary while talking about those themes explicitly name that *"You know, you can […] be a part of something but not really belong. If you just are not active in […] what you're supposedly a part of." (FG1, Speaker 8).* An additional reason for exclusion from the final survey was organizational priority on impacting and improving belonging as described by Themes 1 & 3, thus making assessment of membership less prioritized. Finally, this theme was excluded from survey development due to concerns from the research team that focus group participants were predisposed to consider belonging as akin to membership because of situational bias (an understanding of participants that the topic relates back to the organization of the YMCA).

### Final survey development

We found validated assessments that touched on part of or all the first three themes (see construct mapping of YMCA themes to validated surveys in paragraphs above. Summary of mapping can be found in Table 5 of the Supplementary Materials. The community-academic team came to consensus decision about which validated survey items to incorporate based on concordance with our understanding of the themes, ultimately resulting in six item survey. We included two questions related to value as seen in Theme 1, one question that asked directly about belonging as elevated by Theme 2, one question on loneliness as an antonym of sense of belonging, and two questions that address the relational aspects of belonging noted in Theme 3. The resultant survey can be seen in Fig 1 below:

External member checking with non-focus group members at the YMCA suggested survey item #6 was confusing in its phrasing and thus was excluded. The five remaining belongingness items were included in a larger survey distributed

On a scale of 1-5 (1 strongly disagree, 5 strongly agree), please indicate your agreement with each of the following statements:

1. I feel like I matter to people at the YMCA
2. I feel like I belong to the YMCA
3. I am connected to the YMCA community
4. I feel safe to be my true self at the YMCA
5. I know that I'm not alone because of my community at the YMCA
6. What makes me special as a person is valued by my community at the YMCA

**Fig 1. YMCA Belonging Measurement Tool.** Six item survey assessing sense of belonging as understood by the local YMCA community. Note: item 6 was excluded from final survey after external member checking was completed.

to YMCA membership. The internal consistence of the items was assessed using Cronbach's alpha. This analysis was performed in R 4.2.2, using a bootstrap resampling procedure. Amongst respondents completing the belongingness items ($n = 4318$, $\alpha = 0.868$) there is evidence of good internal consistency amongst the items included to assess belongingness. The bootstrap 95% confidence interval for the Cronbach's alpha [0.861, 0.875] suggests the estimate is stable amongst resamples.

## Discussion

In this study, we contribute to the existing body of belonging research by illustrating a participatory research methods approach to measuring belonging as understood by the community of interest. We additionally add addressing belonging in a new context – a large membership-based organization with membership that spans multiple ages and life stages. We did this through first understanding how our community of interest conceptualizes sense of belonging and then comparing that to the literature to inform the creation of a measurement tool while shortening the time frame typically required of novel survey creation and validation. Guided by the principles of CBPR, which centers community voice within and throughout research processes, we offer a replicable approach to measuring any construct within a community space that spans a variety of demographics. Our discovery includes community concepts that informed the measurement tool and additionally provide insight into ways in which non-academic communities consider belonging in their lives.

When comparing observations in our community context to previously made observations by researchers in the theoretical, education, and employment contexts, we found several similarities. Similarities included belonging being named as the antidote to loneliness; the idea that deep belonging is seated in connection, authenticity, and trust; and the consensus that belonging requires reciprocity and bidirectional engagement. Because these tenets of belonging have already been vigorously studied, it allowed us to paraphrase wording from previously validated questionnaires to assess belonging at the YMCA in a newly compiled survey. [18–27] An additional observation we made, which is congruent with existent literature, is that members spoke of both state (situation-specific) and trait (core psychological need) belonging. [12] Interestingly, they often did so without distinguishing between the two types. This interchangeability of the two types of belonging was seen by members describing belonging as innate and what makes us human while also naming that they felt different amounts of belonging in different spaces. Both in the literature and in this study measurements of belonging are focused on state-specific belongingness. For future work, it is worth considering which type is more impactful on wellbeing and adjusting both measurements and interventions accordingly.

In addition to the similarities we found between the reflections of our study population and the existent literature, there were several new observations that provide insight into context-specific belonging. Firstly, our participants considered belonging as a universally positive experience with a multitude of benefits on both the individual and community level. Belonging researchers, in contrast, have illustrated context-specific negative impacts of belonging. [28–32] Additionally, our focus group participants did not meaningfully distinguish between belonging, connection, and membership although

they are considered separate concepts academically. [12] This lack of distinction could be seen not only when participants named them as synonymous outright but also when participants often did not distinguish between connections formed through deep bonding, trust, and authenticity and those that they formed in an exercise class through simple acknowledgements and nonverbal cues such as smiles. In both situations, members spoke of the feeling of belonging without remarking on the differences between the two. Participants might have been predisposed to think of belonging in the context of their membership to the YMCA, given the study backdrop and purpose. Even so, the lack of distinction provides further evidence that there are context-specific variations in belonging that are worth understanding as we consider the impact of belonging on wellbeing. Taken in total, the nuances of how belonging is conceptualized in our study provides further evidence that belonging is context specific. Community-engaged approaches for understanding and measuring sense of belonging are therefore likely to be more subjectively meaningful and objectively valid than approaches that do not seek to understand context first.

This study does have several limitations. The first is that belonging is closely tied with engagement. [32,33] In our study, we did not recruit members stratified by engagement (to include perspectives from those with low and high engagement at the YMCA). We therefore suspect that we have sampled from a more highly engaged group of members (who presumptively have higher belonging at the YMCA), leading to the lack of voice from members who are poorly engaged (and presumably experience lower belonging at the YMCA). Additionally, all focus groups were held at one branch of the Greater Cincinnati YMCA. Though intentionally selected for its branch diversity, members of other branches may have provided different perspectives on belonging. Another important limitation is that although six themes regarding sense of belonging conceptualization emerged from the focus groups, only three of those themes were captured in existent validated assessments. Although our method of survey development better meets the pace of community improvement work, by not assessing half of the themes we take on significant risk of not accurately measuring sense of belonging in our community. Finally, we caution people against making inferences regarding sense of belonging in the Greater Cincinnati area based on our study. This is for several reasons, including that the community of study was based on organizational membership, and that such membership is inherently limited in several ways (including economically, geographically, and based on transportation abilities). The purpose of our study was not to describe belongingness across the region, and therefore this is not a limitation in and of itself, but rather supports our belief that measurements and interventions in sense of belonging work needs to be community-centered and context specific. In this vein, our methodological approach is easily replicated and broadly generalizable to unique community contexts.

## Conclusion

In this study, we illustrate a process for creating a community-specific measurement tool for sense of belonging in a non-educational, non-employment related organization. We demonstrate similarities and differences to prior literature in how local YMCA members understand belonging. This work is the first step in a larger project that seeks to use survey measurement to inform community-responsive interventions that improve sense of belonging at the YMCA.

## Supporting information

**S1 Fig. Semi-structured interview guide. This guide was used in all focus groups to assess YMCA member conceptualization of belonging.**
(DOCX)

**S1 Table. Belonging focus group participant demographics as compared to YMCA branch demographic makeup.**
Column 1 is the demographic. Column 2 is the percent breakdown at the YMCA branch across all members. Column 3 is the percent breakdown of focus group participants.
(DOCX)

**S2 Table. Mapping of validated surveys and focus group themes to final YMCA questionnaire.** Column 1 shows the phrasing of the question in the YMCA survey. Column 2 demonstrates the scale that we obtained the phrasing from. Column 3 demonstrates the qualitative theme that informed the question.
(DOCX)

**S3 Table. Quote tables. Full quote tables for each theme, with associated codes and quotes.**
(DOCX)

## Acknowledgments

We are grateful for the contribution of time, energy, and thought provided by members & staff at the Clippard Family YMCA in pursuance of this work.

## Author contributions

**Conceptualization:** Kate Hanisian, Carley Riley.

**Data curation:** Monique Quinn.

**Formal analysis:** Monique Quinn, Sierra Roundtree, Lauren Porter, Kimberly Cutler.

**Funding acquisition:** Monique Quinn, Carley Riley.

**Investigation:** Monique Quinn.

**Methodology:** Kate Hanisian, Carley Riley.

**Project administration:** Monique Quinn, Sierra Roundtree.

**Resources:** Monique Quinn, Sierra Roundtree.

**Supervision:** Kate Hanisian, Carley Riley.

**Validation:** Carley Riley.

**Writing – original draft:** Monique Quinn.

**Writing – review & editing:** Sierra Roundtree, Kate Hanisian, Carley Riley.

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
