## [Decision Letter · Decision Letter 0]

19 Oct 2025

PONE-D-25-15800Who belongs? Co-creating an assessment to improve belonging in a community spacePLOS ONE?

Dear Dr. Riley,

We look forward to receiving your revised manuscript.

Kind regards,

Mary Diane Clark, PhD

Academic Editor

PLOS ONE

**Journal Requirements:**

3. Please expand the acronym “NIH” (as indicated in your financial disclosure) so that it states the name of your funders in full.

4. We note that your Data Availability Statement is currently as follows:

“All relevant data are within the manuscript and its Supporting Information files.”

5. We note that this data set consists of interview transcripts. Can you please confirm that all participants gave consent for interview transcript to be published?

If they DID provide consent for these transcripts to be published, please also confirm that the transcripts do not contain any potentially identifying information (or let us know if the participants consented to having their personal details published and made publicly available). We consider the following details to be identifying information:

- Names, nicknames, and initials

- Age more specific than round numbers

- GPS coordinates, physical addresses, IP addresses, email addresses

- Information in small sample sizes (e.g. 40 students from X class in X year at X university)

- Specific dates (e.g. visit dates, interview dates)

- ID numbers

Or, if the participants DID NOT provide consent for these transcripts to be published:

- Provide a de-identified version of the data or excerpts of interview responses

- Provide information regarding how these transcripts can be accessed by researchers who meet the criteria for access to confidential data, including:

a) the grounds for restriction

b) the name of the ethics committee, Institutional Review Board, or third-party organization that is imposing sharing restrictions on the data

c) a non-author, institutional point of contact that is able to field data access queries, in the interest of maintaining long-term data accessibility.

d) Any relevant data set names, URLs, DOIs, etc. that an independent researcher would need in order to request your minimal data set.

For further information on sharing data that contains sensitive participant information, please see: https://journals.plos.org/plosone/s/data-availability#loc-human-research-participant-data-and-other-sensitive-data

If there are ethical, legal, or third-party restrictions upon your dataset, you must provide all of the following details (https://journals.plos.org/plosone/s/data-availability#loc-acceptable-data-access-restrictions):

1. A complete description of the dataset

2. The nature of the restrictions upon the data (ethical, legal, or owned by a third party) and the reasoning behind them

3. The full name of the body imposing the restrictions upon your dataset (ethics committee, institution, data access committee, etc)

4. If the data are owned by a third party, confirmation of whether the authors received any special privileges in accessing the data that other researchers would not have

5. Direct, non-author contact information (preferably email) for the body imposing the restrictions upon the data, to which data access requests can be sent

7. Please include captions for your Supporting Information files at the end of your manuscript, and update any in-text citations to match accordingly. Please see our Supporting Information guidelines for more information: http://journals.plos.org/plosone/s/supporting-information....

8. Please remove your figures from within your manuscript file, leaving only the individual TIFF/EPS image files, uploaded separately. These will be automatically included in the reviewers’ PDF

**Additional Editor Comments:**

Please see my attached comments also. The topic is important and can contribute to how to establish a sense of belonging

As you are aware I truly struggled to get reviewers or this paper so I am acting as the second review and I disagree with the one review that I did get.

It appears that you are doing two different things: one is a qualitative study of people's view of belonging while the other is the development of an assessment. I strongly recommend removing the assessment piece as this time as there is much more work to do to get reliability and vality of an assessment.

The lit review, the method, the analysis all need much more detail--some suggestions in the attached file.

I recommend you access other qualitative articles and see how thematic analysis are conducted. Your results are close but the how you got there needs expanded. You need to expand your own positionally as well.

If you want to focus on the development of the assessment--please look at how other measures/surveys are developed.

Reviewers' comments:

Reviewer's Responses to Questions

**Comments to the Author**

1. Is the manuscript technically sound, and do the data support the conclusions?

Reviewer #1: Yes

2. Has the statistical analysis been performed appropriately and rigorously?

Reviewer #1: Yes

3. Have the authors made all data underlying the findings in their manuscript fully available?

Reviewer #1: Yes

4. Is the manuscript presented in an intelligible fashion and written in standard English?

Reviewer #1: Yes

Reviewer #1: Really nice study and appreciate the authors' emphasis on community-context and community voice, particularly with their explanation that belonging can contribute to health and well-being. Most of my suggestions are surrounding language clarification to ensure the reader fully grasps the points of emphasis by the authors. There are some lines drawn from themes to survey elements that could be further clarified. Additionally, the 3 themes that did not map to validated survey structures seem to be the meat of the authors findings but are not really explored or further measured by the ultimate survey that was created. If individuals were to mimic this study, the authors should give guidance for what groups should do if similarly multiple themes on belonging to not map to current validated studies. I would encourage the authors to push further on the need for more broad survey tool development that better reflects actual community perspectives; particularly if belonging is too be measured as a tool to drive interventions that can improve health.

The authors use "wellbeing" and "well-being" in different part of the manuscript; would be consistent in using one or the other

Line 130  would consider including a citation for CBPR as this is a model paper it can direct readers to learn about this approach and ensure they are true to the method when engaging in this work in their communities; particularly as this will reach an academic audience

Line 142  would change & to "and"

Line 186  given focus on local community context, would further specify location of this YMCA such as by rewording to "This work is a case study examining how Cincinnati YMCA members experience.." or "...how members of a local YMCA experience..."

Lines 202/203  please clarify if it was important for project team members to understand or agree/come to consensus on the themes"

Lines 243-245  "external member checking" is not fully clear in terms of what was done to validate the survey; was it given to members of nearby YMCA branches to complete? did these branches reflect the same community demographics? Why was the same YMCA membership not used through individuals not in the focus group if the goal is in-community construction and utilization? If the nearby branches are considered part of the same community might be of benefit to explain this further. As it reads, not fully clear what measurement validity included.

Lines 259-260  Important information in results so would remove the parenthesis and just state these results.

Line 261-262  suggested rewording here: "Thematic analysis revealed six major themes about belonging (see Supplemental Materials for quote tables)."

Table 2, 3, and 4  slightly unclear if the Survey Item Phrasing are examples from the existent scales or ones that the authors created for their own survey; consider re-naming column "Existent Survey Item Phrasing" to make clear to the reader

Line 305-307 and Table 3  Don't fully follow the logic from belonging being positive/well known to it therefore being asked directly. Can the authors draw this connection out further? (quite different from Theme 1/table 2 and Theme 3/Table 4.

Line 324  "This ___ is fostered through..." -- can the authors clarify what "This" is referring to? maybe rewording as "This high belong is fostered..." or "This ability to be truthful is fostered..."

Line 348 and 350  "This was true..." and "This was similarly seen..." -- can the authors clarify what "This" is referring to in each segment; given multiple statements prior best to be specific as to what the quotations are supporting. consider: "This commonality as a drive of belonging was true for recreational..." and "This expression of shared feelings was similarly seen..."

Lines 368-370  why didn't this information inform the measurement of sense of belonging? Can the authors explain this further? Or why was it necessary for the research team if not part of the measured objective?

Theme 5 and 6  would include something similar to Theme 4 that this theme was not reflected in existent surveys or why it was otherwise not used to generate the community-specific belonging survey by the authors

Line 398 --" again use of "This" without clarity on what it is referencing; suggestion "This connection between belonging and group member was seen when participants..."

Line 442  Not sure if the focused-on themes were nuanced and evolving as they were reflected in existing validated tools? Maybe the approach to create a focused survey reflects the specific community perspectives and values. The 3 themes not reflected are very interesting in terms of "evolving" ways that belonging is described and viewed.

Line 447 & 454 & 467 " again would clarify what "This" in the sentence "This included belonging..." is referring to and what "This was seen by..." is referring to and what "This could be..." is referring to

line 452  keep study description/findings in past tense for consistency; "additional observation we make" to "we made"

*not including themes 4-6 as a study limitation?? leaving out whole aspects of this community's definition or expression of belonging

For the Literature Review, was there a timeframe for the search or any articles from any year past were included?

Is the community project partner being trained by the study lead a possible limitation of the study given these were the 2 who then did the initial theme evaluation? Not sure what is standard for qualitative projects to this may not have any bearing.

.

Reviewer #1: No

---

## [Author Response · Author response to Decision Letter 1]

22 Jan 2026

We provide a comprehensive response in our submission.

---

## [Decision Letter · Decision Letter 1]

23 Feb 2026

Who belongs? Co-creating an assessment to measure belonging in a community space

PLOS One

Dear Dr. Riley,

This revision is much clearer and addresses most of my concerns from the earlier review. Please check the COREQ checklist to be sure to highlight that in your revision. Thank you for the clarifications.

Here are a few minor things to address:

A sense of belonging –need an article

Avoid hyphens

A combination of literature review,and extraction, and qualitative research methodsextraction, and qualitative research methodsextraction, and qualitative research methodsextraction, and qualitative research methods

Wellbeing is well-established (need citations when you say this)

191 is not literature review—take out that heading

This is the finding of items for your measure—give it a better heading

Concluding sentences after direct quotes to end a paragraph

No contractions outside of quotes

Line 380 not sure what is happening with that table and then going into the next theme—same as Table 3.

We look forward to receiving your revised manuscript.

Kind regards,

Mary Diane Clark, PhD

Academic Editor

PLOS One

Journal Requirements:

Additional Editor Comments:

This version of your manuscript is much clearer. You need to do a few things and then I will be the final reviewer to determine if these items are clear.

The journal requires this information so please clarify and make thee items obvious:

Use the COREQ checklist which is on the Plose One Website to highlight the following 6 issues:

1--defined objective/research question

2--description of sampling strategy, rational for recruitment, inclusion and exculsion prcedures, and # of partcipants

3--Data Collection Procedures

4--data analyses procedure to enable replication

5--potential bias

6--limitations

Most of these items are in your mansucript but check and make sure that they are obvious.

Reviewers' comments:

Reviewer's Responses to Questions

**Comments to the Author**

Reviewer #2: All comments have been addressed

2. Is the manuscript technically sound, and do the data support the conclusions?

Reviewer #2: Yes

3. Has the statistical analysis been performed appropriately and rigorously?

Reviewer #2: Yes

4. Have the authors made all data underlying the findings in their manuscript fully available?

Reviewer #2: Yes

5. Is the manuscript presented in an intelligible fashion and written in standard English?

Reviewer #2: Yes

Reviewer #2: The manuscript is well prepared and addresses the topic clearly. For future research, the author(s) may wish to consider expanding the number of participants to further strengthen the study and enhance the generalizability of the findings.

.

Reviewer #2: No

---

## [Author Response · Author response to Decision Letter 2]

8 Mar 2026

Thank you for the opportunity to further improve the manuscript. We provide a response to reviewer and editor comments in our updated submission.

---

## [Editor Report · Decision Letter 2]

11 Mar 2026

Who belongs? Co-creating an assessment to measure belonging in a community space

PONE-D-25-15800R2

Dear Dr. Riley,

We’re pleased to inform you that your manuscript has been judged scientifically suitable for publication and will be formally accepted for publication once it meets all outstanding technical requirements.

Kind regards,

Mary Diane Clark, PhD

Academic Editor

PLOS One

Additional Editor Comments (optional):

Thank you for these changes. The mansucript is clear and follows the CORE formatting. Congratulations on an interesting article.
---

## [Editor Report · Acceptance letter]

PONE-D-25-15800R2

PLOS One

Dear Dr. Riley,

I'm pleased to inform you that your manuscript has been deemed suitable for publication in PLOS One. Congratulations! Your manuscript is now being handed over to our production team.

Kind regards,

on behalf of

Dr. Mary Diane Clark

Academic Editor

PLOS One